# Study of the Effects of Initial Cutting Conditions and Transition Period on Ultimate Tool Life when Machining Inconel 718

**DOI:** 10.3390/ma14030592

**Published:** 2021-01-27

**Authors:** Morvarid Memarianpour, Seyed Ali Niknam, Sylvain Turenne, Marek Balazinski

**Affiliations:** 1Mechanical Engineering Department, Polytechnique Montreal, Montreal, QC H3T 1J4, Canada; morvarid.memarianpour@polymtl.ca (M.M.); sylvain.turenne@polymtl.ca (S.T.); marek.balazinski@polymtl.ca (M.B.); 2Sustainable Manufacturing Systems Research Laboratory, School of Mechanical Engineering, Iran University of Science and Technology, Tehran 1684613114, Iran

**Keywords:** tool wear, Inconel 718, turning, built-up edge

## Abstract

Rapid tool wear and limited tool life are major problems when machining Inconel 718, which still need further attention. Amongst the reported strategies, limited studies have been reported on optimizing initial cutting conditions by means of tool life improvement. Therefore, in this work, the tool wear progress and tool life were investigated by varying the initial conditions in the transition period, which was set at four seconds. The transition point was discovered by previous works by the authors. After the transition point, similar cutting conditions were used as the reference condition. The tool wear morphology and size were recorded and analyzed in each condition. It was revealed that applying a lower cutting speed and feed rate in the transition period led to improved tool life as compared to the reference condition. In other words, the use of optimum levels of cutting parameters in the transition period of the cutting process may enhance tool life at higher cutting time. For instance, initial feed rate (0.15 mm/rev) and cutting speed (25 m/min) led to the improvement in the ultimate tool life by about 67% and 50%, respectively. Besides, applying the lower initial cutting speed, i.e., 25 m/min, increased the tool life by about 50% when the insert reached the maximum flank wear (*v*_Bmax_) of 300 µm in comparison with those at higher initial cutting speeds. This phenomenon may lead to better insight into the effect of the influence of the initial cutting conditions in the transition period on tool life when machining hard-to-cut materials. Moreover, the built-up edge (BUE) was exhibited as the primary wear mode in all cutting conditions.

## 1. Introduction

Among hard-to-cut superalloys, Inconel 718 has a wide variety of real-world applications in numerous industrial products and sectors [1,2,3]. As a high-temperature material, Inconel 718 is widely utilized in aerospace applications. When needed for high-performance materials with promising mechanical as well as thermal properties (i.e., high strength and oxidation-resistant materials), Inconel 718 is considered as a specifically appropriate alternative for use in structures of aero engines and gas turbines, where temperature is high [4]. Despite wide applications and prominent features, the poor machinability of Inconel 718 at precision machining conditions is still considered a subject of interest, and additional investigations on optimizing cutting parameters and specifying appropriate cutting tools are still needed. Amongst machinability attributes, limited studies were reported in the literature about tool wear and tool life improvement when machining superalloys, including titanium metal matrix composites (Ti-MMC) and Inconel 718.

As is well understood in the literature, the tool wear index is generally referred to as the maximum permitted flank wear (*v*_Bmax_) that can be measured in the course of the cutting process, and the corresponding results can be observed in the tool wear curve [5]. According to Figure 1, three typical sections exist in the tool wear curve. As noted in [6], the main wear modes are initial or running-in period (I), steady wear within the steady-state (II), and accelerated wear (III) [7,8]. It has been known that both initial and steady wear periods occur very fast in the machining of superalloys, including Inconel 718. To prolong the tool life, one solution is to assess the wear modes and morphologies in both states and employ the adequate cutting parameters and lubrication strategies to reduce the presence of wear modes aforementioned. To that end, several studies were conducted to understand the wear modes in the steady-state wear period [9,10]. The capability of three well-known cutting tool materials including PVD-TiAlN coated carbides, Al2O3-TiC ceramic, and CBN in the turning of Inconel 718 was studied by Xavier et al. [11] with various cutting speeds (60, 80, and 120 min), feed rates (0.08, 0.10, and 0.12 mm/rev), and depth of cuts (0.2, 0.4, and 0.6 mm). It was revealed that the most suitable cutting speed for carbide inserts is 90 m/min at which the minimum value of the maximum *v*_Bmax_ was achieved. On the other hand, Xavier et al. [11] investigated the *v*_Bmax_ in three different cutting tools including PVD TiAlN coated carbide, Al_2_O_3_–TiC ceramic, and CBN tools, and revealed that the optimal conditions for dry-cutting, MQL, and flood cooling are: cutting speed of 90 m/min, feed rate of 0.16 mm/rev, and depth of cut of 0.4 mm [12].

Among all three wear periods mentioned earlier, the effects of machining parameters on the steady-state wear period have been thoroughly investigated. Still, adequate knowledge on the wear modes and morphology, as well as factors governing elevated initial tool wear morphology and size in machining hard to cut materials, are still demanded in wider scopes. In this regards, having a precise insight into the initial tool wear mechanism, morphology, and influencing parameters on initial tool wear size may result in better control and optimization of the machining process, and thus, extending tool life when machining hard-to-cut materials such as Ti-MMC [13,14,15,16]. Therefore, within the first seconds of the machining operation, it is strongly recommended to study the initial tool wear mechanisms. In this venue, Duong et al. [8] mentioned certain cutting conditions that influence the initial tool wear and tool life while turning Ti-MMCs. It was mentioned that the chaos theory could be employed to describe the relationship between initial tool wear and ultimate tool life. Applying the optimal values of cutting parameters, including cutting speed (*v*_c_), depth of cut (*a*_p_), and feed rate (*f*_r_) within the very first seconds of machining, may result in the improvement of the tool life. Memarianpour et al. [17] studied the effects of cutting conditions, with specific emphasis on cutting speed (*v*_c_) and lubrication modes on tool wear progression when turning Inconel 718.

To identify and monitor the predominant tool wear mechanisms, several advanced characterization techniques, including scanning electron microscopy (SEM), energy-dispersive X-ray spectroscopy (EDX), and X-ray photoelectron spectroscopy (XPS) were conducted on the cutting tools and inserts [18]. Abrasion, adhesion, and diffusion were considered as the most prevalent tool wear mechanisms occurring under different conditions while machining nickel-based alloys [19,20]. One other key element in machining superalloys is the right selection of cutting tools. Commercial cutting tools such as polycrystalline diamond (PCD), cubic boron nitride (CBN), and physical vapor deposition (PVD) carbide, as well as ceramic, are among the best choices for machining hard-to-cut materials, including titanium alloys Ti-6Al-4V and MMC. Several studies reported the machining of superalloys using ceramic tools [21].

Furthermore, the progress of carbide tools wear in turning Inconel 718 was also investigated. In general, the PCD and CBN tools show a lower degree of wear rate, while the coated carbide tools are more economical choices [8]. However, for machining superalloys such as Inconel 718 at relatively low cutting speeds, the PVD carbide tools are the primary choice [21,22,23,24,25,26,27,28]. However, the initial tool wear mechanism of carbide tools in high-speed machining needs to be further examined in wider scopes at different values of cutting conditions and cutting time. The lack of knowledge aforementioned was partially studied by Memarianpour et al. [17]. Unfortunately, no sufficient source of data has been provided about the variation of *v*_Bmax_ when machining Inconel 718 using different cutting tools. 

In most of the works published in the literature, the tool wear was evaluated based on the measured values of flank wear (*v*_Bmax_) in the course of cutting time [5], and no additional work was found on the effects of initial tool wear morphology and mechanism on ultimate tool life when machining Inconel 718. Therefore, within this study, the effects of initial cutting conditions on the ultimate tool life were studied in turning Inconel 718 with carbide inserts.

Furthermore, this work intends to present the effects of initial cutting parameters on the ultimate tool life when machining Inconel 718. The proposed method can efficiently enhance the service life of the cutting tool. Moreover, to align with green manufacturing goals and objectives, dry machining was used to omit lubricants utilization. In general, the proposed method can be easily implemented in the industrial and manufacturing sectors. 

## 2. Experimental Work

### 2.1. Experimental Plan

The workpiece used was a cylindrical Inconel 718 with the nominal bulk hardness and tensile strength of about 37.0±0.63 (R_C_) and 1100 MPa, respectively. This was delivered in the shape of a cylindrical bar with a diameter of 114 mm and the chemical compositions based on the weight percent of elements (wt %) listed in Table 1. The cutting tool used in all experiments was carbide/cermet inserts CNMG 432-MR4 TS2500 coated with TiCN/Al_2_O_3_ (CVD), which was fabricated by Seco, Inc. (Figure 2a, Montreal, Canada). The CNC-MAZAK Nexus 200 (MAZAK, Montreal, Canada) was applied in the turning tests (Figure 2b). It ought to be noted that to prevent the coolant effects, experimental tests were undertaken under dry conditions. The experimental works were performed twice, and the average values of the obtained *v*_B_ were utilized in successive studies. A new and sharp insert edge was used in each machining test. The following cutting angles were used in the experimental tests: rake angle *γ*_0_ = −6°, back rake angle *γ*_p_ = −6°, nose radius *R*_ϵ_ = 1/32 mm, tool cutting angle *k*_r_ = 95°, and insert thickness *s* = 3/16 mm.

As noted earlier, according to the research outcomes reported in previous works by authors [17,29], the cutting time 4s was considered as the transition point, and the first four seconds of cutting operations was considered as the transition period, and the five different cutting conditions, as listed in Table 2, were used in the transition period. The first cutting condition was considered the reference condition, and it was employed after the transition point in all the experiments. In this venue, the cutting conditions used after the transition point were similar to the reference condition. 

All the experiments have been repeated twice to respect repeatability concerns. 

### 2.2. Tool Wear Measurement

The experiments were carried out in two steps. The first stage was devoted to investigating the tool life during the machining of Inconel 718. In this regard, cutting speed and feed rate at 45 m/min and 0.25 mm/rev were kept constant, respectively. At the following stage, initial cutting speeds, as well as different initial feed rates ranging from 25–65 m/min and 0.15–0.35 mm/rev were used, respectively. It ought to be noted that the depth of cut was kept constant at 1 mm in all the experiments. 

To study insert wear mechanisms, size, and morphology, as well as elemental analysis and quantitative mapping, two SEM (JEOL, JSM-840A and JEOL JSM 7800F FEG-SEM, St-Hubert, Canada) equipped with an Oxford X-ray detection system (AZtec EDS, St-Hubert, Canada) were used. Moreover, Jeol JSM 7800F is equipped with field emission guns (FEG), which provides an extreme resolution of 0.8 nm at 15 kV and 1.2 nm at 1 kV. The micrographs of the microstructures were obtained at both low and high magnifications. The maximum values of recorded flank wear measurements were considered for additional studies. 

## 3. Result and Discussion

### 3.1. Effects of Cutting Parameters on Tool Life 

Generally, cutting tool material undertakes severe mechanical and thermal stress when machining superalloys. These phenomena could appear as a result of high cutting stress and temperature near the cutting edge, which affects the cutting tool wear rate and short tool life [30,31,32]. As noted earlier, the cutting speed is considered as one of the most crucial parameters that directly affect tool life. Moreover, tool life analysis in turning with variable feed rates when machining Inconel 718 was not reported in the literature. Therefore, in order to remedy the lack of knowledge observed, it was intended to investigate the effects of cutting parameters and initial wear modes on the initial and ultimate wear modes and tool life machining of Inconel 718. Therefore, according to Table 2, besides the reference cutting condition, four different strategies with a combination of various levels of feed rate and cutting speed were used. In this venue, the flank wear values in the transition period (0 < *t* < 4s) as well as uniform conditions (*t* > 4 s) were presented as a function of cutting time (Figure 3). The individual recorded values of *v*_B_ were also shown in Table 3. As can be inferred from Figure 3 and Table 3, the use of higher levels of cutting speed (*v*_c_ = 65 m/min; condition 3) in the transition period led to higher flank wear (*v*_Bmax_). Thus lower tool life was recorded as compared to the reference condition at higher cutting time. In contrary, based on Table 3 and Figure 2, using the lower cutting speed (*v*_c_ = 25 m/min; condition 2) led to lower *v*_Bmax_ at 4 s. Thus, better tool life was also recorded at higher cutting time. Feed rate is also another key factor that affects the wear morphologies and *v*_Bmax_. Therefore, two different feed rate strategies, as noted as condition 4 and condition 5 (Table 2 and Table 3), were used at constant cutting speed (45 m/min) and depth of cut (1 mm). 

As can be seen in Table 3 and Figure 3, the use of a lower feed rate (condition 4) at the transition period led to better tool life as compared to the reference condition (condition 1), while less tool life was recorded when the higher feed rate was used in the transition period (condition 5). In other words, increasing the feed rate from 0.15 to 0.35 mm/rev in the transition period led to increased flank wear (*v*_Bmax_) from 111.0 to 126.0 μm at the end of the transition period (*t* = 4 s), respectively. Similarly, a higher cutting time, more rapid tool wear was shown. Knowing that a *v*_Bmax_ of 300 μm is considered as the maximum permitted tool wear size in most of the reported works, ultimately, the permitted cutting time at the *v*_Bmax_ of 300 μm was dramatically decreased from 38.5 to 15.1 s when the initial feed rate was changed from 0.15 to 0.35 mm/rev, respectively. Similarly, compared with reference mode (condition 1), the permitted cutting time at the *v*_Bmax_ 300 μm was dramatically decreased from 33.9 to 17.4 s when the cutting speed was changed from 25 m/min to 65 m/min. 

In principle, according to reported works in the literature [17,33], it can be exhibited that the combination of lower levels of feed rate and cutting speed may improve tool life. The main reason could be attributed to less presence and effects of friction, heat generation, and micro-structural deflections in the first few seconds of machining at lower levels of cutting parameters. However, these phenomena could be reconfirmed upon the adequate experimental characterization of the tool wear morphologies, vibration monitoring, and heat generation monitoring within the transition period. 

The experimental results in Figure 3 and Table 3 also confirm that incorporating small adjustments in cutting conditions at the initial periods of cut may tend to make a massive difference in tool life at higher cutting time. According to experimental results, a direct relationship can be formulated between the initial tool wear and ultimate tool life. This relationship is widely affected by cutting parameters such as *v*_c_, *a*_p_, and *f*_r_. This phenomenon agrees with the principles of chaos theory, which stated a negligible difference at the beginning of the process may tend to large differences at a higher time. At relatively low cutting speeds, only the mechanically activated wear mechanisms occur (e.g., BUE formation, adhesion, abrasion, etc.). However, increasing cutting speeds and thus cutting temperatures may lead to thermal wear mechanisms (diffusion, oxidation, etc.). In other words, heating the cutting tool results in losing its strength, activating the diffusion, and chemical wear mechanisms [19].

### 3.2. Experimental Characterization of Wear Morphology 

Figure 4 and Figure 5 depict the SEM and EDX images for the wear mechanisms observed at different conditions. The wear mechanisms observed were mainly adhesive, abrasive, and BUE on the tool flank face. The abrasive wear is a result of the presence of hard particles and impurities within the workpiece material [34], such as carbon, nitride, and oxide compounds, as well as built-up fragments. The adhering material became a stable BUE protecting the face. The adhesive wear is due to the high temperature and pressure during cutting, which causes micro welds between the clean and fresh surfaces of the chip and the rake face [3].

The analysis and assessment of wear morphologies and mechanisms were conducted from SEM and EDX techniques. As shown in Figure 4, slight BUE formations were observed on the SEM images of the cutting tools within the transition period. It can be noted that the BUE formation was exclusively observed in the cutting processes with the initial cutting speed of 45 m/min (Figure 4, conditions 1), while almost no BUE was detected when the initial cutting speeds of 25 and 65 m/min were employed (Figure 4, conditions 2 and 3). On the other hand, the adhesion mechanism was observed in all inserts (Figure 4, conditions 1–3). Besides, as shown in Figure 4, the flank wear observed in all three workpieces increased by higher initial cutting speed. These results further confirmed that the use of an initial cutting speed 25 m/min resulted in lower flank wear (Figure 4, condition 2). Furthermore, upon applying the lower initial cutting speed (i.e., *v*_c_ = 25 m/min), almost double cutting time was resulted at *v*_Bmax_ of 300 µm as compared to the other tested conditions. In other words, lower cutting speed led to 50% improvement in the tool life as compared to conditions 1 and 3 with higher cutting speed (Figure 3). Furthermore, the SEM analysis of cutting tools in steady-state at *t* = 26 s (Figure 5) revealed that all the cutting tools experienced severe BUE formation and adhesion. However, BUE formation was much more severe under conditions 1 and 3 in comparison with findings related to condition 2. Abrasion was also noticed in the tests. However, due to lack of space, only EDX analysis of BUE formations was presented in this work.

On the other hand, at *t* = 4 s, the use of a lower initial feed rate (0.15 mm/rev, condition 4) and constant cutting speed (*v*_c_ = 45 m/min) led to less *v*_Bmax_ (Figure 4, condition 4) as compared with those strategies with higher initial feed rates (Figure 4, conditions 1 and 5). Moreover, as expected, slight BUE formation was also observed when a higher initial feed rate was used (Figure 4, conditions 1 and 5). Besides, at constant cutting speed (45 m/min) and higher initial feed rate, decreased tool life by about 67% was resulted (Figure 3). In other words, the lower the initial feed rate applied, the higher the tool life achieved. In addition, a great extent of BUE formation, adhesion, and abrasion can be observed in the steady-state mode (*t* = 26 s), which can be observed in Figure 5 under conditions 1, 4, and 5. However, the degree of BUE formation was noticeably higher under conditions 1 and 5 in comparison with that under condition 4.

To achieve a better insight into the wear mechanisms on the cutting tools, EDX analysis was conducted, and the associated results were presented in Figure 4 and Figure 5. The obtained data from EDX analysis of the cutting tool in the transition time (*t* = 4 s) at the constant feeding rate of 0.25 mm/rev and the initial cutting speed of 25 m/min revealed that the coating layer of the cutting tool was almost preserved (Figure 4, condition 2). In other words, the degree of adhesion and diffusion were negligible when lower initial cutting speed was used. Besides, the peaks corresponded to various phases of Ni, Cr, and the other major elements existing in the structure of workpieces were observed in EDX results, indicating that adhesion occurred in the transition period when the initial cutting speed was adjusted to 25 m/min (Figure 4, condition 2). Similar observations were made when the initial feed rate of 0.15 mm/rev at constant cutting speed of 45 m/min was applied (Figure 4, condition 4). Additionally, through the obtained EDX results, one might conclude that the coating layers (mainly consisting of Ti, Al, etc.) of cutting tools were slightly damaged and/or completely removed under all conditions. In other words, the peaks related to Ti and Al elements were omitted at the end of the transition period under all conditions, while the peaks attributed to Ni, Cr, Fe, etc., appeared.

The EDX results obtained at steady state (*t* = 26 s) reveal that the cutting tools used in the cutting process under conditions 1, 3, and 5 lost their external coating layers. The presence of peaks corresponding to different phases of Ni, Cr, Fe implied the effectiveness of the adhesion mechanism, as mentioned earlier. The BUE formation, adhesion, and abrasion occurred with a lower degree when the cutting process was carried out under conditions 2 and 4. Ultimately, in the case of performing the cutting process under condition 5, abrasion mechanisms were also observed due to the presence of the peak corresponding to W (Figure 5, condition 5).

## 4. Conclusions

The literature review emphasizes that inadequate studies are available about factors governing initial tool wear morphology and size (*v*_Bmax_) when machining Inconel 718. No work was found on investigating the correlation between the appeared initial tool wear within the transition period and ultimate tool life when machining hard-to-cut materials, in particular, Inconel 718. Therefore, in this work, the tool wear progress and tool life were investigated based on varying the initial conditions at the transition period, set at four seconds. As noted earlier, the transition point was discovered from the previous works by the authors. After the transition point, similar cutting conditions as the reference condition were used.

The obtained experimental results revealed that the cutting speed was the most effective cutting parameter to the flank wear size, followed by feed rate. 

Both adhesion and abrasion wear was observed during the cutting process via the various mechanisms attributed to the thermo-mechanical effects. This observation can be stated in terms of an increment in the cutting force and temperature, resulting in micro scales welds formation at the tool-workpiece interfaces. However, such phenomena could be improved by applying lower initial cutting speed and lower initial feed rate.

The transition period was defined in this work, and the turning tests with various cutting conditions were conducted. Cutting parameters were carefully chosen in compliance with the instructions made by the manufacturer of cutting tool, Seco Inc.

Five different cutting conditions were used in the initial period of cut, the so-called transition period. After the transition point 4s, similar cutting conditions as the reference condition was undertaken within the specific time intervals up to 56 seconds. The tool wear morphology and size were recorded and analyzed in each case.

The following conclusions may be taken from the findings of this study:Less initial tool wear was observed under a lower cutting speed (*v*_c_ = 25 m/min) and feed rate (*f*_r_ = 0.15 mm/rev).In most cases, BUE and adhesion were predominant wear mechanisms in the transition period (*t* < 4 s).In all cases, the BUE was exhibited as the major wear mode in cutting conditions at the steady-state, while both adhesion and abrasion were observed. Such observations could be attributed to the severe interactions between the hard particles of the Inconel 718 material and the cutting tool.When the *v*_Bmax_ of 300 μm at constant cutting speed (45 m/min) was achieved, increasing the initial feed rate decreased the tool life by about 67%.Applying the lower initial cutting speed, i.e., 25 m/min, increased the tool life by about 50% when the insert reached the *v*_Bmax_ of 300 µm in comparison with those at higher initial cutting speeds.In general, it could be concluded that using optimum levels of cutting parameters in the transition period of cutting process may prolong tool life.

The findings in this work can shed light on the investigation of the effect of the initial condition and initial behavior on tool life for further development and progress. Besides, the results obtained in this work can pave the way for new cutting strategies with particular emphasis on tool life improvements in wider industrial sectors and processes. 

## Figures and Tables

**Figure 1 materials-14-00592-f001:**
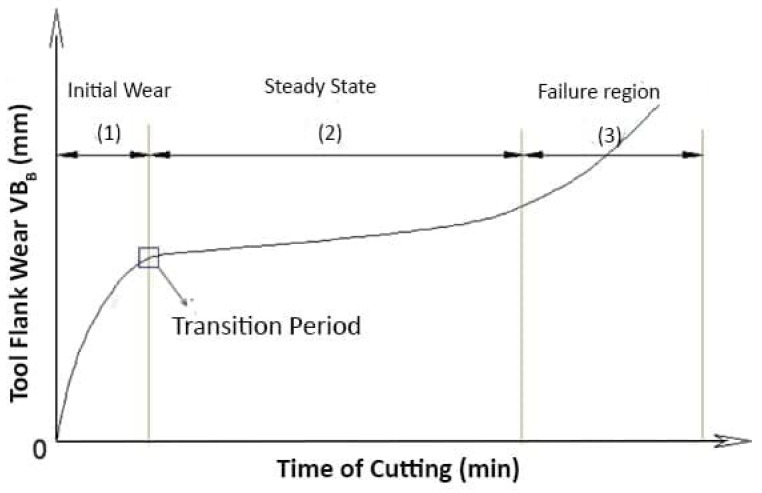
Typical tool wear curve.

**Figure 2 materials-14-00592-f002:**
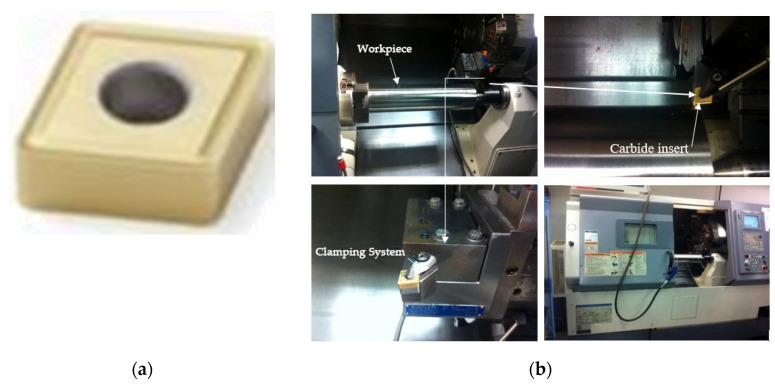
(**a**) The insert used; (**b**) Experimental set-up used.

**Figure 3 materials-14-00592-f003:**
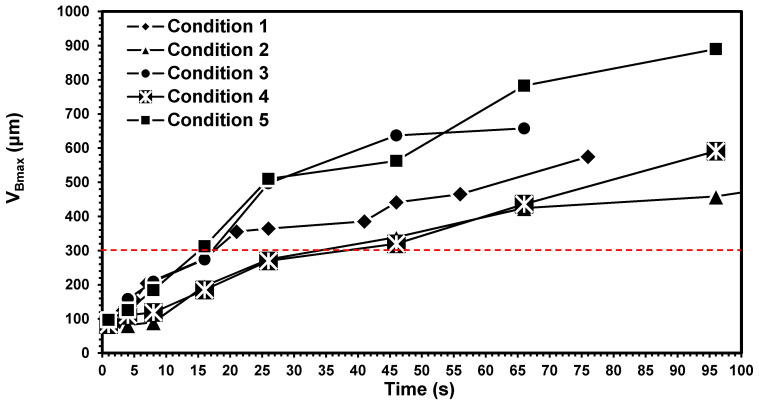
The *v*_Bmax_ vs. time under different cutting strategies.

**Figure 4 materials-14-00592-f004:**
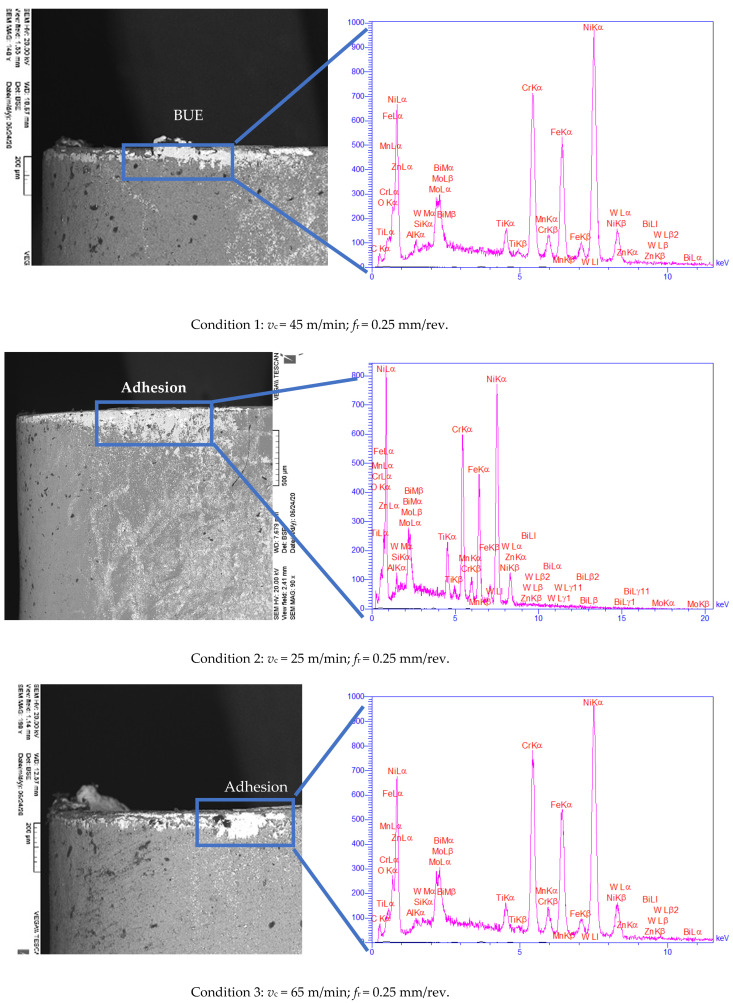
Initial tool wear mechanism on the flank face at *t* = 4 s under various cutting conditions.

**Figure 5 materials-14-00592-f005:**
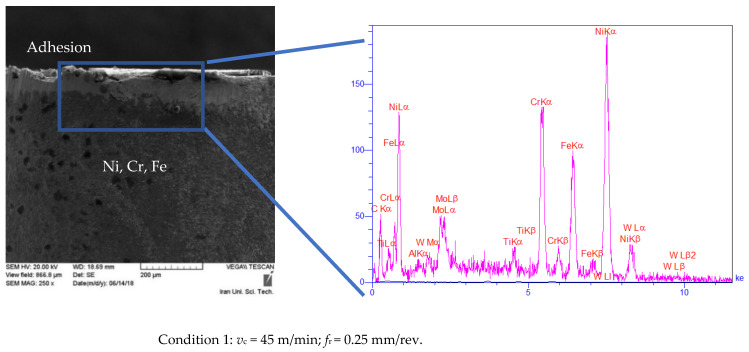
Initial tool wear mechanism on the flank face at *t* = 26s under various cutting conditions.

**Table 1 materials-14-00592-t001:** Chemical composition of Inconel 718 alloy (wt %).

Element	(wt %)	Element	(wt %)
Ni	53.4	Si	0.12
Cr	18.8	Mn	0.07
Cb	5.27	Cu	0.07
Mo	2.99	C	0.03
Ti	1.02	P	0.01
Al	0.50	Fe	Balance
Co	0.17		

**Table 2 materials-14-00592-t002:** Experimental conditions used.

Cutting Conditions	Transition Period (0–4 s)		Cutting Conditions after 4 s	
	*v*_c_ (m/min)	*f*_r_ (mm/rev)	*v*_c_ (m/min)	*f*_r_ (mm/rev)
1	45	0.25	45	0.25
2	25	0.25		
3	65	0.25		
4	45	0.15		
5	45	0.35		

**Table 3 materials-14-00592-t003:** The recorded value of *v*_Bmax_ at each cutting condition.

Cutting Time (s)	*v*_Bmax_ (mm)				
	Cutting Condition 1	Cutting Condition 2	Cutting Condition 3	Cutting Condition 4	Cutting Condition 5
4	0.0994	0.0817	0.1583	0.1110	0.1260
8	0.2045	0.0905	0.2092	0.1190	0.1846
16	0.2756	0.1964	0.2741	0.1850	0.3127
26	0.3644	0.2745	0.4970	0.2700	0.5100
46	0.4412	0.3393	0.6373	0.3200	0.5627

## Data Availability

Data sharing not applicable.

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
