# Peer review of "Study of the Effects of Initial Cutting Conditions and Transition Period on Ultimate Tool Life when Machining Inconel 718"

_materials, 2021, doi:10.3390/ma14030592_

Round 1

Reviewer 1 Report

Dear authors,

The influence of initial cutting conditions on the tool life and tool wear morphology when machining Inconel 718 with a carbide tool is studied in this paper. The paper deals with tool wear problems during the initial period often met in machining operations, but not sufficiently taken into consideration in the literature. In general, the paper is written and structured well although there are some critical mistakes that must be corrected.

The abstract is a good summary to introduce the topic and conclusions drawn in this work. The introduction gives a good idea of the state of the art with 28 references cited throughout the text, including 15 references in the last 10 years, but only 3 in the last three years. In the reviewer’s opinion, the references could be updated to collect more recent works related to the topic. The experimental tests are well described, and the results and conclusions are clear. However, I detected some typographical errors and incongruences within the document that must be addressed before publication. I also have some questions regarding the methodology presented in this paper that may be interesting to include. More precisely, my remarks and my comments are:

Abstract

  1. “The similar cutting conditions as the reference condition were used after the transition point.” After the transition point, the cutting conditions used are the same as the reference condition not similar.

Nomenclature

  1. In my opinion, this section is not necessary since there is not a large number of abbreviations and variables used in the text. I would remove it and follow the recommendations of the journal: Abbreviations should be defined in parentheses the first time they appear in the abstract, main text, and in figure or table captions and used consistently thereafter.”
  2. I would consider representing symbols for quantities in italic in all the document, such as cutting speed as Vc

Introduction

  1. L12-13 “including Ti-MMC” MMC should be defined before using it. The definition of MMC is at L45, correct it.
  2. L29 “eventually may lead to extended tool life [12-15] when machining hard to cut materials such as 29 Inconel 718.” 12-15 analyse the tool life when machining Ti-MMC, not Inconel 718. I would rewrite the sentence to make clear that the studies referenced are related to similar hard to cut metals, but not Inconel 718.
  3. L34 & L36. The cutting speed symbol is defined twice. Correct it.
  4. L44 “PCD, CBN and PVD”. These abbreviations are not defined in the main text. Please define them.
  5. L58 “Therefore, within this study, the effects of cutting parameters on the initial 58 tool wear morphology and ultimate tool life were conducted in turning Inconel 718 with carbide 59 inserts.” I do not understand these sentences, I think that the sentence is not correct, and it should be rewritten for the sake of clarity.

Experimental work

  1. L73 “sharp insert’s edge” The apostrophe is not correct.
  2. L78 “the cutting time 4 s was considered as the transition point” Can it be assumed that the transition point is constant for different cutting conditions? Have the authors tried the sensitivity of the transition point?
  3. L81 “The first cutting condition with similar values of feed rate and cutting speed in both 81 transition and uniform period was considered as the reference condition. Similar cutting conditions 82 were then followed after the transition time in the other four cutting trials.” These sentences are quite confusing, please rewrite them. I understand that the first cutting condition is taken as the reference conditions and it is employed after the transition point in all the tests. In this way, the cutting conditions used after the transition point are the same as the reference condition.
  4. Figure 2a. The quality of the picture is very low.
  5. Table 1. In the caption is referred to wt% and in the table to w%. In addition, the symbol is not defined in the main test.

Results and discussion

  1. L127 “111.2 to 128.9 μm” The values referred do not agree with the values shown in Table 3: 111.0 and 126.0 μm
  2. L146 The variables defined in this line are already defined in the text previously.
  3. Figure 3 Some of the results of table 3 does not agree with the points represented in Figure 3. For instance, Condition 3 at t = 26s VBmax=0.2741 mm and in the graph is around 0.500 mm. In addition, some of the points are not plotted because they are out of the plotting range. Lastly, the markers of the legend should be resized.
  4. SEM of Figure 4 Condition 4 is missing
  5. I guess that is missing and E in the abbreviation “BU” of Figure 5.
  6. L174 “However, BUE 174 formation was much severe under the conditions 1&3 in comparison with findings related to 175 condition 2.” In Figure 5, I can see that condition 1 shows a lower BUE than condition 2. In addition, condition 3 does not show BUE at t=4 in Figure 4.
  7. L161-210 I think that including a table with the main wear modes observed at the transition point and during the steady wear period would help the reader to follow the results and discussion.

References

  1. L258 [3]. The reference is incomplete. Publisher?

Author Response

Please find the responses made to the editorial comments.

Regards

SA Niknam 

Reviewer 2 Report

The Paper is only a technical report and paper not contains a contribution to the scientific area/discipline. This paper not yet reached the level to publish. Below are comments concerning this paper:

1) Abstract: it is somehow simplistic. It should be a very short summary of the paper expressing the novelty. The authors should add two sentences concerning the results themselves.

2) The keywords are missing: built-up edge.  I suggest that the Authors remove it: Initial tool wear.

3) The nomenclature was not made according to the standard. Feed rate is determined by f, maximum flank wear - VBmax (ISO 3685:1993), insert nose radius rε etc. Additionally, Authors should also add units for individual variables in the nomenclature. Some of the abbreviations that can be found in the text are missing, e.g. BUE.

4) In the introduction, the presentation of previous relevant works is poor. The authors mention papers in bulk, e.g. [1-5] and [12-15], without explaining what is important for the analysis presented here. The authors need to make a thorough review of the previous work and explain what is new presented in this paper. Do not give knowledge known for more than half a century (book knowledge) about the wear curve (Figure 1). This is a scientific article that is intended to express novelty.

5) The discussion of the results is poor. The authors try to describe what it can be seen on the graphs without any real explanation of the results. They make assumptions that are not backed-up by any references, provide the results without any theory and as a result, the conclusions are rather trivial. For the first 4 seconds the cutting parameters were changed. How do the authors explain the differences for conditions number 4? First, the VBmax value is about 0.1110 mm (almost the highest value) after 4 seconds and if the tool then worked under the same conditions as the other tools, the wear was the lowest (0.32 mm). No commentary whatsoever.

6) What does the red dashed line in fig. 4 mean?

7) On what basis did the Authors determine the different types of tool wear in SEM images?

8) Conclusions are somehow simplistic as they seems to be observational without revealing findings of generic academic value. Conclusion 6 is completely incomprehensible from the reader's point of view. When the authors used the MQL and MQCL methods, they mentioned these methods in their conclusions.

9) All variables in the article, please write in italic style.

Author Response

(The authors gave the same response as above.)

Reviewer 3 Report

Manuscript ID: materials-968808
Title of Article: „Study the effects of initial cutting conditions and transition period on the ultimate tool life when machining Inconel 718“.

Authors: Morvarid Memarianpour, Seyed Ali Niknam *, Sylvain Turenne, Marek Balazinski

The work is relevant because it analyzes important machining processes that determine machining quality, price, productivity.

It is obvious that the lower parameters used during the tool "run-in" period (VC 25 m / min and fr 0.15 mm / rev) gives lower performance, and at the same time the heating dynamics of the tool. Therefore, a steady increase in feed and cutting speed is obviously appropriate for use. I think it is appropriate to summarize as the volume of metal cut during the "run-in" period.

Comments, shortcomings:

It is not explained what a BUE is / Add to nomenklature.

What is BU??? (fig. 5, 2 and 5 condition) / Edit illustrations (Fig. 5)

It is desirable to include a VB-Flank wear measurement scheme / Sketch.

Repetition of experiments and dissemination of results not clear (Required).

It is unclear whether the maximum experiment duration (46 s) is approximately equal to the actual tool run time. If not equal, why?

With what accuracy and equipment the transition period was measured - 4 s.

The manuscript is not prepared correctly - fig. Figure 4 Condition 4 is placed on different pages. Condition 2 and 5 SEM illustrations are mounted in reverse (mirror principle).

What is the diameter of the workpiece? Specify.

The mechanical properties (hardness, strength) of the workpiece Inconel 718 and the tool must be specified

3 cutting conditions. When working at higher speeds, higher tool wear is "programmed".

FIG. 3. The result is incorrect because the measurements of tools with different parameters of Flank wear were performed at different tool machining intervals, so it is obvious that the tools had different temperature regime and wear at different speeds.

Fig. 3. - certificates of working conditions are incorrectly small (it is difficult for the reader not to make a mistake when analyzing).

Fig. 4 shows that in all cases, after 4 s of performance, the TiCN / Al2O3 coating is already damaged. What is the thickness of the coating on the cutting tool plate? So, what is reason to show that picture???

3 table and figure 3 data are different. On the basis of what data is the 3 figure formed? 3 figure. What is the purpose of the red line?

Conclusion 1. This can be done without an experiment.

The second part of conclusion 6 without studies is not substantiated: under various lubrication strategies, including wet, minimum quantity lubrication (MQL), and minimum quantity cooling lubrication (MQCL)”.

Author Response

Dear Editor

Please find the responses made to the editorial comments.

Regards

SA Niknam 

Reviewer 4 Report

Reviewed article is very interesting and write at high scientific level. Presentation method is good and in accordance with generally accepted standards in that area. Figures, tables as well as terminology are mostly clear and precise. Described method was correctly verified and compared with standard approach to this problem. Below are listed some substantive remarks that should be taken into consideration by the Authors to improve reviewed text:

  • at the end of the introduction should be clearly and concise given the research gap to create the appropriate lead up for the motivation of the work;
  • the novelty of given approach should be emphasized in introduction;
  • I suggest to provide more precise information about used experimental and measurement positions,
  • I suggest also to give wider description of potential use of presented findings in scientific research as well as in industrial practice;
  • the strengths and limitations of the obtained results and applied methods should be clearly described;
  • in conclusions deeper explanation of observed phenomena and trends should be given (conclusions should refer not only to results but also to causes of obtained results);
  • the conclusions should highlight the novelty and contribution to the state of the knowledge in given area.

After a careful study of the text sent for review, many editorial comments also come to mind:

  • all mathematical/physical symbols should be write italics and proper subscript/superscript notation for better readability of the text,
  • figure cannot be larger than the page (fig. 5) – figure and its caption should be in the same page,
  • consequently all values should be writing with space before its unit (with very few exceptions),
  • capital V (volume) is not a proper designation of velocity (small v).

Author Response

(The authors gave the same response as above.)

Round 2

Reviewer 1 Report

The authors have answered satisfactorily all my questions and suggestions. I have no more comments. 

Author Response

Dear reviewer #1

Please accept our warm appreciation for your comments. 

Regards

S Ali Niknam, Ph.D

Reviewer 3 Report

Manuscript ID: materials-968808
Title of Article: „Study the effects of initial cutting conditions and transition period on the ultimate tool life when machining Inconel 718“.

Authors: Morvarid Memarianpour, Seyed Ali Niknam *, Sylvain Turenne, Marek Balazinski

The work is relevant because it analyzes important machining processes that determine machining quality, price, productivity.

I see progress in the article

115-116. The workpiece was a cylindrical Inconel 718 with the nominal bulk hardness and strength of about 37.0 ± 0.63 (RC) and ……???, respectively. What is the strength of the material??? 

FIG. 3 (1 Condition). The result is incorrect because the measurements of tools with different parameters of Flank wear were performed at different tool machining intervals, so it is obvious that the tools had different temperature regime and wear at different speeds.

What is the structural significance of the EDX analysis of the cutting edge (Fig. 4 and Fig. 5), if it is not related to structural differences in the analysis of the tool wear values.

Author Response

Dear Editor and reviewer

Please accept our warm appreciation for your comments. The responses to your comments were listed as follows, and those comments with required amendments in the manuscript were addressed and the changes were highlighted in yellow.

Regards

S Ali Niknam, Ph.D

Reviewer 4 Report

All my comments were taken into acount in revised version of the manuscript.

Author Response

Dear reviewer #4

Please accept our warm appreciation for your comments. 

Regards

S Ali Niknam, Ph.D
